# A New Urban Functional Zone-Based Climate Zoning System for Urban Temperature Study

**Zhaowu Yu** [1], **Yongcai Jing** [2], **Gaoyuan Yang** [3] **and Ranhao Sun** [2,*]

1   Department of Environmental Science and Engineering, Fudan University, Shanghai 400438, China;
    zhaowu_yu@fudan.edu.cn
2   Research Center for Eco-Environmental Sciences, Chinese Academy of Sciences, Beijing 100085, China;
    ycjing_st@rcees.ac.cn
3   Department of Geosciences and Natural Resource Management, Faculty of Science,
    University of Copenhagen, 1958 Copenhagen, Denmark; gy@ign.ku.dk
*   Correspondence: rhsun@rcees.ac.cn

**Abstract:** The urban heat island (UHI) effect has been recognized as one of the most significant terrestrial surface climate-related consequences of urbanization. However, the traditional definition of the urban–rural (UR) division and the newly established local climate zone (LCZ) classification for UHI and urban climate studies do not adequately express the pattern and intensity of UHI. Moreover, these definitions of UHI find it hard to capture the human activity-induced anthropogenic heat that is highly correlated with urban functional zones (UFZ). Therefore, in this study, with a comparison (theory, technology, and application) of the previous definition (UR and LCZ) of UHI and integration of computer programming technology, social sensing, and remote sensing, we develop a new urban functional zone-based urban temperature zoning system (UFZC). The UFZC system is generally a social-based, planning-oriented, and data-driven classification system associated with the urban function and temperature; it can also be effectively used in city management (e.g., urban planning and energy saving). Moreover, in the Beijing case, we tested the UFZC system and preliminarily analyzed the land surface temperature (LST) difference patterns and causes of the 11 UFZC types. We found that, compared to other UFZCs, the PGZ (perseveration green zone)-UFZC has the lowest LST, while the CBZ (center business district zone)-UFZC and GCZ (general commercial zone)-UFZC contribute the most and stable heat sources. This implies that reducing the heat generated by the function of commercial (and industrial) activities is an effective measure to reduce the UHI effect. We also proposed that multi-source temperature datasets with a high spatiotemporal resolution are needed to obtain more accurate results; thus providing more accurate recommendations for mitigating UHI effects. In short, as a new and finer urban temperature zoning system, although UFZC is not intended to supplant the UR and LCZ classifications, it can facilitate more detailed and coupled urban climate studies.

**Keywords:** urban temperature; urban functional zone; big data and cloud computing; point of interest; urban-functional-zone-based climate zone; city management

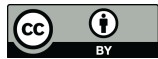

## 1. Introduction

The urban climate is one of the hot topics in urban environment research [1]. According to the World Urbanization Prospects 2018, 55% of the world's population were urban dwellers in 2018, and by 2050, that number is expected to reach 68%. Urbanization has transformed natural surfaces into a coupled human and natural system, which alter the materials, energy flows, radiation, and composition of the atmospheric structure in the near-surface layer [2–5]. During urbanization, the modification of terrestrial surfaces, changes in surface material and the albedo of buildings (pavements), the geometry of the surface structure (e.g., spacing and height of buildings), and anthropogenic heat emissions cause one of the most significant impacts on the Earth's surface climate—the urban heat island (UHI) [6–10].

In general, the quantification of UHI is related to the urban canopy layer, which is observed using thermometers to measure the air temperature ($T_a$) near the ground ($T_a$ can be measured in a weather screen or ventilated radiation shield, at one or more sites considered to be representative of urban and rural areas); air in this layer is typically warmer than that at screen height in the countryside [1]. Hence, the classical definition of the urban–rural classification (UR) of UHI is that the atmospheric temperature in the urban area is warmer than its surrounding rural areas [6,11–13]. The UR classification of UHI has given researchers and decision-makers a simple and intuitive framework to separate the urban and rural effects on local climate [14]. However, the classical UR classification has recently suffered critical challenges [15–18]. For instance, Stewart and Oke [14] suggested that while the definitions of urban and rural may be evocative of the landscape, they are vague as an object of scientific analysis. It also claimed that more than three-quarters of the observational UHI literature fails to give the local or micro-scale character of those sites and rarely reports on the site metadata necessary to quantify or otherwise elucidate the terms 'urban' and 'rural' [19]. In particular, the classical UR classification is becoming outmoded today, especially in the densely populated (i.e., Asia) developing urban agglomeration regions [12]. It also mentioned that the relationship between urban and rural is a dynamic and continuous process rather than a dichotomy [20], while the spatial demarcation between the urban and rural is normally artificial, the term urban has no single, objective meaning and thus has no climatological relevance as Stewart and Oke [14] stated. Therefore, they proposed a new classification schema (local climate zone, LCZ) to facilitate consistent and climatologically relevant studies and classifications. The schema includes 17 standard classes at the local scale ($10^2$ to $10^4$ m), and each class is unique in its combination of surface structure, fabric, and metabolism (to some extent). However, Although LCZ supplements the deficiencies of UR classification, it also has many limitations, such as being hard to apply in cities with complex land use (i.e., Jakarta, Mumbai, Manila) and urban agglomerations (which will be discussed later). For instance, Ren et al. (2016) [21] investigated two big Chinese cities Wuhan and Hangzhou by LCZ schema, and they found that the LCZ map does not correspond with the real conditions. They also noted that "*The existing LCZ classification scheme has been developed based on the experiences and observations made in European and US cities. It may not be adequate to describe the land features of cities in China.*" These statements clearly prove the limitations of the LCZ schema.

Furthermore, previous research [12] also indicated that classical UR classification is not suitable to examine the heat effect in the context of urban agglomerations, and the concept of UHI should be replaced by region heat island (RHI). Moreover, there have also been other classifications to describe urban climate (temperature) in cities like urban terrain zones (UTZs) [22], urban climate zones (UCZs) [2], urban zones for characterizing energy partitioning (UZEs) [23], and Climatopes [24].

However, cities are highly populated areas with various socioeconomic activities. Hence, by contrast with the Earth's surface transformation that affects urban temperature by altering the sensible and latent heat fluxes [25–27], the human activity-induced anthropogenic heat is a direct external source to the urban thermal environment, particularly in high-density neighborhoods at micro-local spatial scales [28–30]. However, previous definitions (i.e., UR and LCZ) on UHI find it hard to capture the anthropogenic heat. On a larger scale, a recent study in China even found over the last decades that the contribution of fossil fuel $CO_2$ to urban temperature has become greater than that from land-cover and land-use change [31]. Anthropogenic heat can greatly affect the urban (micro) climate and spatial-temporal variations of UHI within a city [30]. On the other hand, anthropogenic heat is highly correlated with the urban functional zones (UFZ), and the concept of UFZs includes the nature and socio-economic properties of urban heat and microclimate variability [30,32]. In general, UFZ is segmented by urban road networks and is the basic unit of quantitative analysis in urban refinement planning and management [33]. As Yuan, et al. [34] said: "modern cities develop with the gestation, formation, and maturity of different functional

zones; and these functional zones provide people with various urban functions to meet their different needs of socio-economic activities." The UFZ is normally characterized by similar spectral features, socio-economic function, and structured by a specific function and, therefore, has a similar energy consumption and the outdoor thermal environment [32]. The urban temperature studies thus also need to pay more attention to the roles of human activities (UFZ-based) rather than a physical property dominated climate-based classification, such as the LCZ system (although LCZ also mentioned the effect of human activities).

Nevertheless, one of the critical challenges in this topic is that the accuracy of UFZ mapping is limited. Specifically, at present, most functional zone datasets derive from field surveys, which is time-consuming and hard to update at a later point. Fortunately, with the development of remote sensing (RS) and social sensing (i.e., social media data, public transport check-in/out data, point of interest (POI) and location-based service (LBS)) technology recently, some previous studies suggested that effective and scientific combination of the two data sources allows us to better understand the human activities and UFZ patterns [35,36]. However, studies combining multi-source data, such as RS and SS (social sensing) data to mapping UFZ still lacking [33,37], which limits the understanding of the effects of UFZ on UHI patterns. Moreover, with the development of urbanization, urban agglomeration has become the most salient feature of global urbanization in recent decades [12,38,39]; hence the simple functions and structures of cities turn into multifunctional and complex structures. These require us to examine the urban thermal environment from a more comprehensive perspective, and UFZ is an important carrier for relevant analysis.

Therefore, to address the aforementioned insufficiencies and provide a general microscale UHI definition system, the study aims to: (1) propose a new method to mapping UFZ with the employment of RS and SS technology; and (2) further put forward a new urban functional zone-based urban temperature (UFZC) zoning system for urban temperature studies; as well as combining the advantages (and limitations) of UFZC classification with previous classification schemas (UR and LCZ schema). The proposed UFZ mapping method and particularly the UFZC would deepen our understanding of the urban thermal environment at a finer scale.

## 2. Methodology

### 2.1. Identification Framework of Urban Functional Zone-Based Urban Temperature (UFZC)

Big data and cloud computing have injected new vitality into urban climate (temperature) research. For instance, massive mobile phone positioning data, POI data, and land use-type polygon data all can be crawled from Google Map (or Alibaba's Amap) and Google Earth Engine (GEE, a cloud computing platform) [40]. Therefore, the UFZC system we proposed integrates big data and cloud computing technologies, as well as remote-sensing technology.

As mentioned above, UFZ is defined as an area with similar socioeconomic and physical properties [32,33]. The UFZs are usually cut into sub-regions by street networks; therefore, street network preparation is the priority. We obtained the street network in two different ways in this framework. One is from *OpenStreetMap* (https://www.openstreetmap.org/). Another one is from Amap Custom Map Mode (https://lbs.amap.com/dev/mapstyle/edit?styleid=yourself key), which can crawl colorful raster data by python and selenium code on the website after getting the free key code. Supplemental Material A shows the Python programming code (PPC) to obtain the street networks. It needs to be mentioned that the *OpenStreetMap* is a useful way to obtain the street network, but the process of cutting sub-regions regularly is very cumbersome. Therefore, we used the ArcScan which is a part of ArcMap toolboxes to convert colorful raster data to polygon ShapeFile.

The detailed technical process to obtain the UFZC is shown in Figure 1. Firstly, we use the Python script to get the street from *OpenStreetMap*. However, the street network is too confusing to divide the study area into sub-regions and it takes a long time to process the street network. Therefore, we developed a practical *Online Map Data Crawling System* (*OMDCS*) (Supplemental Material B), which has a complete and convenient function for

obtaining the street network. The ***OMDCS*** can be used to download rich urban data and land-use attributes data such as commercial areas, transportation hubs, etc., which are more detailed than the results of remote-sensing images. Secondly, since the agricultural area, rural residential areas, and land surface temperature (we used LST to represent the urban temperature in this test, see below) cannot be downloaded through the ***OMDCS***, we therefore obtained these data through remote-sensing images. Thirdly, we used ***Google Earth Engine*** (https://code.earthengine.google.com/) to retrieve the agricultural field and suburban residence. GEE data were provided by Landsat-8 images, which were acquired on 28 December 2017, consistent with the POI acquisition date.

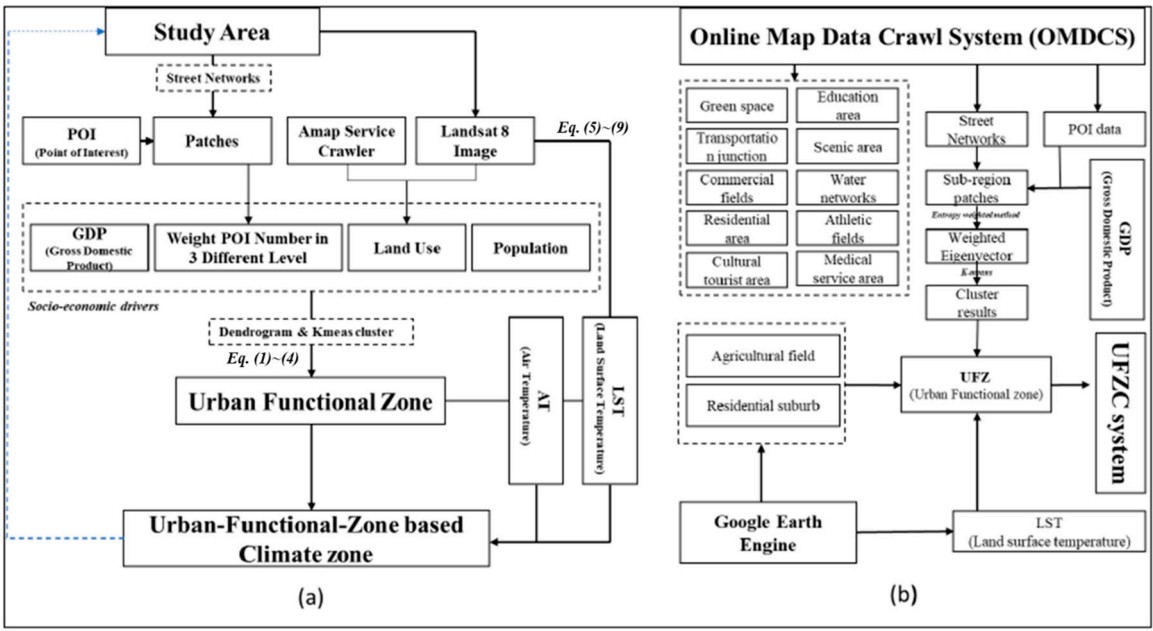

**Figure 1.** The technical framework of urban functional zone-based urban temperature (UFZC) system identification. (**a**) is the general technical framework of UFZC system identification; (**b**) is the specific technical process to obtain the UFZ and UFZC system.

### 2.2. Point of Interest (POI) Data Processes

We used the Python Programming Code (Supplemental Material A) to crawl the POI data. The data crawl target was Amap (https://lbs.amap.com/api/webservice/guide/api/search), which is the most widely used local LBS (location-based service) provider in China. POI data is a new type of effective spatial data that can truly reflect social and economic activities and is therefore an effective way to identify urban functional zones [37]. In general, the POI data are broken down into 23 types, including Recreation, Catering industry, Automotive Services, Financial, Education, Public, Health Care Services, Hospitality, Residence, Organizations, and Travel, etc. Twenty of these types are stable categories, and the other 3 categories are real-time incidents such as traffic accidents and road maintenance incident. Each POI has 6 column properties: name, coordinates, and categories in 3 hierarchy levels are composed of division, group, and class that we called level 1, level 2, and level 3, respectively. The weights of the categories in the 3 levels are different. For example, level 1—Hospitality Service, including star-rated hotel, budget hotel, and Inn that can distinguish the functional property of the sub-regions. Hence, the standardization of data must be done at 3 levels, respectively.

Each sub-region patch can be depicted by an eigenvector consisting of 3 levels of POI type and gross domestic product (GDP) of the district (county), so each sub-region is an Eigenmatrix that we can perform cluster analysis. Since the difference between the amount of maximum and minimum POI in the 3 levels of all sub-region patches is varied,

and the importance of the number and type of POIs at the 3 levels are different. Therefore, a new data standardization method is needed to determine the functional characteristics of sub-regions more scientifically and efficiently. The correlation of each of the above metrics is sophisticated and uncertain, the entropy method (EA) can capture implicit interactions between the factors and indicate the level of each metric [41–43]. More importantly, we can determine the weight of the comprehensive evaluation of the 3 levels of POI type and obtain the last weighted eigenvector of the sub-region patch.

*Step 1*: The POI data will be standardized in level 3 according to the following methods. Each sub-area patch is a matrix of the number of POI categories:

$$y_{ij} = \frac{x_{ij} - min\{x_j\}}{max\{x_j\} - min\{x_j\}} \tag{1}$$

where $y_{ij}$ is the standardized matrix data, $min(x_j)/344\ max(xj)$ is the minimum and maximum values, $x_{ij}$ is the amount of POI category in level 3.

*Step 2:* Calculate the entropy of the POI type in level 3. The entropy of information is an important factor to measure the weight of evaluation metrics. The large entropy of information indicates that the information provided by the metrics in the comprehensive score is large and the weight is high. The equation below indicates how to calculate the entropy of information.

$$E_j = -ln(n)^{-1} \sum_{i=1}^{n} p_{ij} ln(p_{ij}) \tag{2}$$

$$p_{ij} = \frac{y_{ij}}{\sum_{i=1}^{n} y_{ij}} \tag{3}$$

where $n$ is the amount of the POI data on one dimension, $y_{ij}$ is the standardized data, and suppose when $y_{ij} = 0$, $p_{ij} = 0$.

*Step 3:* Calculate the weight of different types of POIs in one dimension in level 3. After calculating the information entropy, the entropy theory is used to determine the weight of each category in level 3, which reflects the importance of subcategories in the evaluation system.

$$W_i = \frac{1 - E_i}{k - \sum E_i} (i = 1, 2, \ldots, k) \tag{4}$$

where $k$ is a constant, $k = 1/ln(m)$, $m$ is the amount of the sub-region.

*Step 4*: Repeat the same process for Level 2 and Level 1 based on the results of the previous. Then we can obtain the weighted amount of POI in level 1 of each sub-region.

*Step 5*: Regarding the weight amount of POI of each sub-region and GDP data, we can cluster the sub-region patches by *K-means* and *Dendrogram Cluster methods*. Finally, we can obtain the similarity among sub-region patches.

*Step 6:* Based on the land-use polygon shapefile and POI 363 features, we identified the UFZs.

According to urban planning and management practices as well as the previous studies [30,32,34], we divided the UFZ into 11 types. It needs to be mentioned that these 11 types are basic UFZs. Based on research needs and data accuracy, these 11 types of UFZ can be divided into several sub-types, such as high-intensity residence zone and low residence zone.

### 2.3. Land Surface Temperature Acquisition of UFZs

The UFZC system includes 11 UFZ-based urban climate zones, and the corresponding temperature can be obtained in a variety of ways, such as weather station (2 m height) air temperature [15], boundary layer air temperature [2], and land surface temperature (LST) [3,9,27]. In the following section, the case study, due to the data limitation, the Landsat-8 Operational Land Imager and Thermal Infrared Sensor (OLI_TIRS)

remote-sensing images of Beijing were used to represent urban temperature and test the UFZC system.

Previous studies have demonstrated that LST retrieved by the radiative transfer equation (RTE) algorithm can obtain the highest LST accuracy in high atmospheric water vapor environments [12,27,44]. Hence the RTE algorithm proposed by Jiménez-Muñoz, et al. [45] is selected to calculate LST in this test case. The equation can be expressed as the apparent radiance received by a sensor:

$$T_s = \frac{K_2}{\ln\left(\frac{K_1}{B(T_s)} + 1\right)} \tag{5}$$

where $T_s$ is the LST (K), $K_1$ is 774.885 W m$^{-2}$ sr$^{-1}$ μm$^{-1}$, $K_2$ is 1321.079K, $B(T_s)$ is the ground radiance. According to Plank's law, $B_i(T_s)$ can be expressed as:

$$B(T_s) = \frac{L_\lambda - L_\uparrow - \tau(1 - \varepsilon)L_\downarrow}{\tau\varepsilon} \tag{6}$$

$$L_\lambda = gain \times DN + bias \tag{7}$$

where gain and bias are the gain and bias value for band 10 from the Landsat metadata file respectively, $L_\lambda$ is the radiance value from DN (digital number) value by radiance calibration, $L_\uparrow$ is the upwelling path radiance ($L_\uparrow$ = 0.26 W m$^{-2}$ sr$^{-1}$ μm$^{-1}$), $L_\downarrow$ is downwelling path radiance ($L_\downarrow$ = 0.48 W m$^{-2}$ sr$^{-1}$ μm$^{-1}$), $\tau$ is atmospheric transmittance ($\tau$ = 0.96), $\varepsilon$ is land surface emissivity calculated based on the single-channel method which is different in water, artificial surface and nature surface, and it can be expressed as [46,47]:

$$\begin{cases} \varepsilon_{\text{water}} &= 0.995 \\ \varepsilon_{\text{artificial}} &= 0.9589 + 0.086P_v - 0.0671P_v^2 \\ \varepsilon_{\text{nature}} &= 0.9625 + 0.0614P_v - 0.046P_v^2 \end{cases} \tag{8}$$

where $P_v$ is fraction of vegetation cover, vegetation coverage, expressed as [48,49]:

$$P_v = \left(\frac{NDVI - NDVI_{\text{soil}}}{NDVI_{\text{veg}} - NDVI_{\text{soil}}}\right)^2 \tag{9}$$

where $NDVI$ is the normalized difference vegetation index (NDVI) of the mixed pixel, $NDVI_{\text{soil}}$ is the NDVI of bare soil and $NDVI_{\text{veg}}$ is the $NDVI$ of the vegetation.

The key software or tools used in this study including ArcGIS Desktop [50], Python [51] and PyCharm IDE [52]. There are some of the core Python packages that have been used including NumPy, SciPy, pandas, matplotlib. The entropy method was also undertaken based the entropy method equations in Python.

Finally, after the above process, we can perform UFZC classification in any target area (Beijing).

## 3. Case Study

### 3.1. Study Area

Beijing (39°26′–41°30′N, 115°25′–117°30′E) is the capital city of China. It has an area of 16,808 km$^2$, including 14 districts and two counties, and in 2015 the permanent population reached 21.7 million [53]. Beijing has a typical continental monsoon climate with four distinct seasons, hot and rainy summers, and cold and dry winters. The annual average temperature in Beijing is 12.3 °C, and the annual precipitation is about 572 mm [54]. Since the late 1980s, Beijing has experienced a rapid and disorderly sprawling urbanization process, resulting in many urban environmental problems and UHI effects [55,56]. The spatial pattern of development in Beijing is a typical concentric expansion, showing a ring-shaped pattern with distance from the city center to the outskirts [57]. With rapid urbanization, the UFZ of Beijing has become diverse and mixed. According to the latest administrative divisions, from a broader perspective, Beijing can be divided into four func-

tional areas [53]: (1) core functional zone; (2) urban function extended zone; (3) new urban development zone; (4) ecological conservation zone. In general, as an urban metropolitan area, Beijing covers almost all types of urban function, so it is an ideal case area to test the UFZC system.

### 3.2. Data Source and Processing

According to the technical framework of the UFZC determination (Figure 1), selecting Beijing as the case area, we first used the Python script to obtain the street network of Beijing from **OpenStreetMap**. As mentioned above, we developed a practical **OMDCS** (Supplemental Material A) (Figure 1b) to improve the efficiency of processing street networks and more detailed land-use attributes data. The details are shown in Table 1 and Figure 2. Secondly, with the help of GEE, we obtained the agricultural area, rural residential area, and LST from remote-sensing images of Beijing (Figure 3). Thirdly, we used Python programming code to crawl the POI data (Supplemental Material A) in Amap and obtained 1,125,472 POI data in December 2017 (Figure 4). After that, we quantified 11 UFZs in Beijing through POI data filtering, standardization, and previous steps (Table 2 and Figure 5). Finally, the corresponding LST-based temperature has been used to represent the urban temperature and form the UFZC system (Figure 6).

**Table 1.** The data crawl by the online map data crawl systems (**OMDCS**).

| Data Type | Description |
| --- | --- |
| Green space | The urban green space, including city parks, greenbelts, residential green, perseveration green. |
| Education area | Research Institutes, Universities, vocational schools, junior-senior high schools, elementary schools. |
| Residential quarter | Residential quarters. |
| Transportation junction | Airports, railway stations, coach stations. |
| Scenic area | City parks, historical sites, cultural and natural scenic resorts. |
| Cultural tourist area | Cultural scenic resorts |
| Medical and health service area | General hospitals, specialist clinics, community hospitals. |
| Athletic fields | Basketball fields, golf courses, football fields, gym, and sports clubs. |
| Commercial fields | Shopping mall, furniture markets, commodity wholesale market centers, etc. |
| Water networks | Urban water, lakes, reservoirs. |
| Buildings | Building boundary polygons. |
| Street networks | Expressways, ring-roads, trunk roads, and other level roads. |

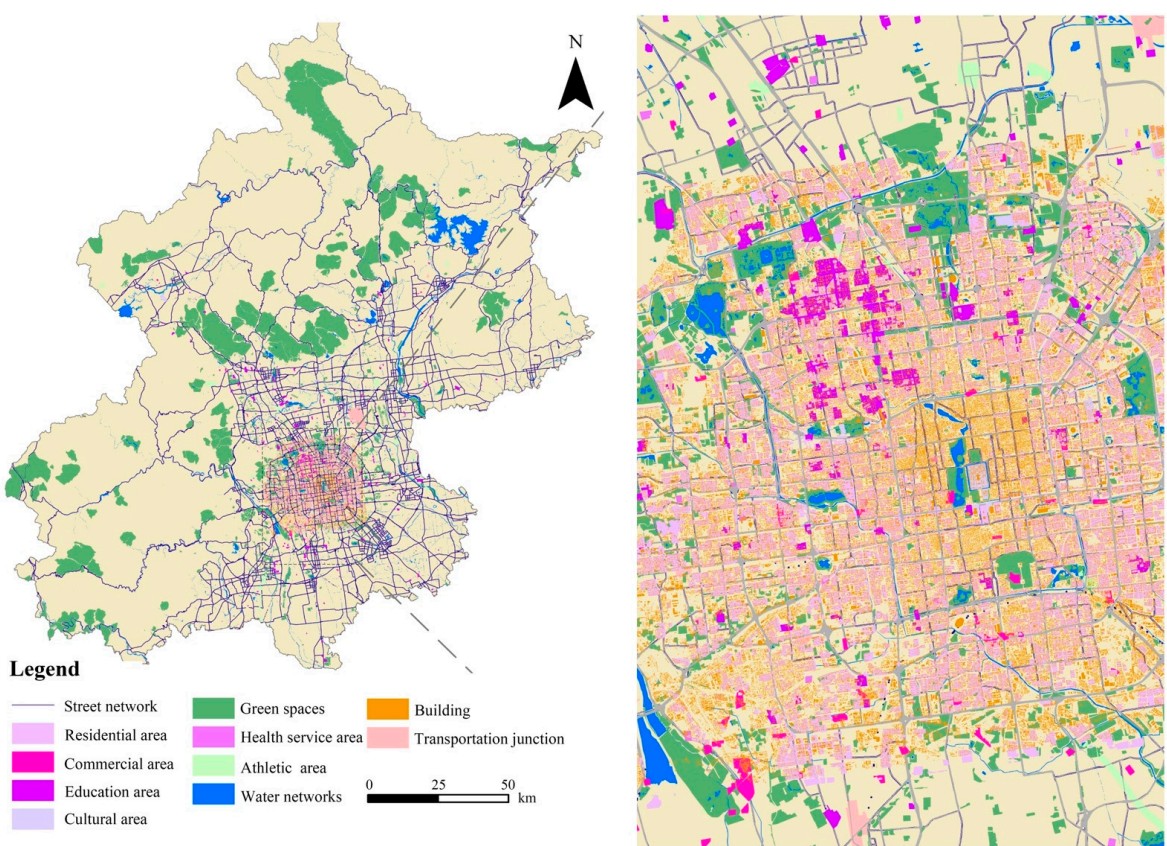

**Figure 2.** Land-use types (**left**) and an enlarged view inside the 5th ring-road of Beijing (**right**).

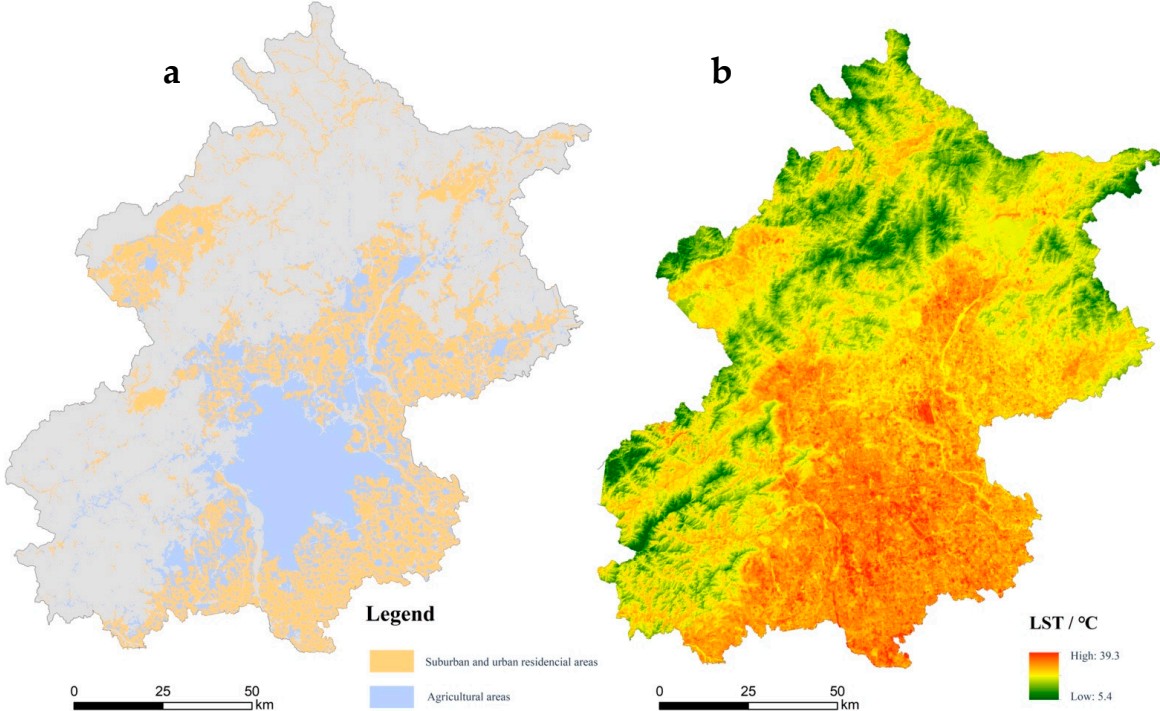

**Figure 3.** Agricultural fields and residence area identification (**a**); and land surface temperature (LST) map (**b**).

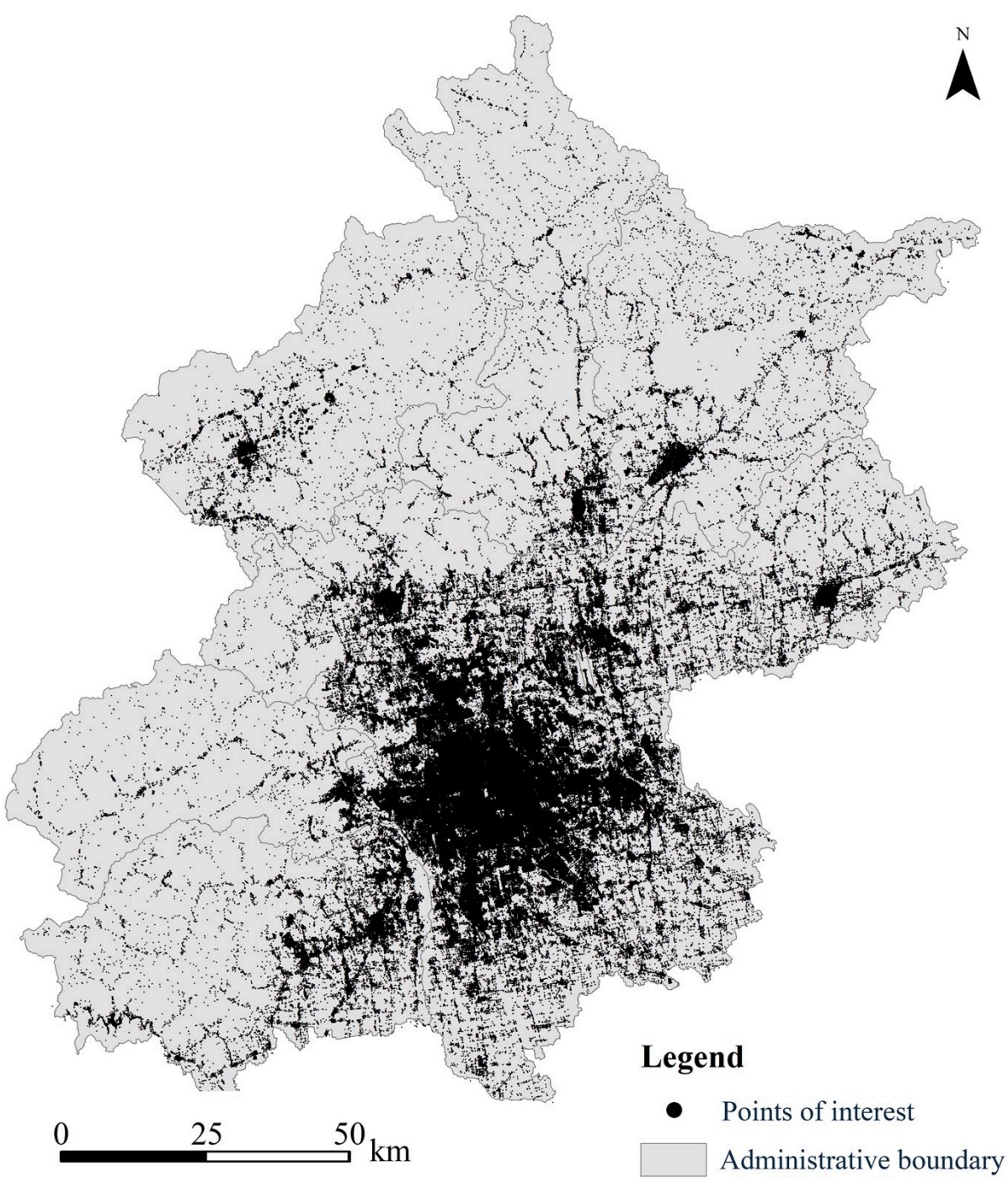

**Figure 4.** The point of interest (POI) data of Beijing.

**Table 2.** Urban functional zone acquisition and interpretation.

| Urban Functional Zone | Abbreviation | Explanation of Division |
|---|---|---|
| Residence Zone | REZ | Impervious, construction material; typical urban communities including multiple family houses and high buildings. |
| Campus Zone | CPZ | Areas for schools, colleges, institutes, government, hospitals, embassies, military bases, etc. |
| Center Business District Zone | CBZ | The concentration of commercial and business. Such as headquarters of insurance, banking, and software companies. It is normally located in the city center. |
| General Commercial Zone | GCZ | General commercial activities, such as shops, hotels, wholesale markets, etc. |
| Agricultural Zone | AGZ | Crops, gardens, and other herbaceous vegetation. |
| Industrial Zone | IDZ | The concentration of factories, workshop, and warehouses. |
| City Water Zone | CWZ | All areas of open water, including rivers, reservoirs, and lakes. |
| Recreation Green Zone | RGZ | Urban parks, golf courses, soccer fields, and other recreation areas. |
| Preservation Green Zone | PGZ | Successional distribution of trees, shrubs, and brushes, such as shelter-forest, isolation belt, urban forest, etc. Natural and manmade grassland. |
| Public Zone | PBZ | City large-scale square, airports, railway stations, coach stations. |
| Main Road Zone | MRZ | Streets, main roads, etc. |

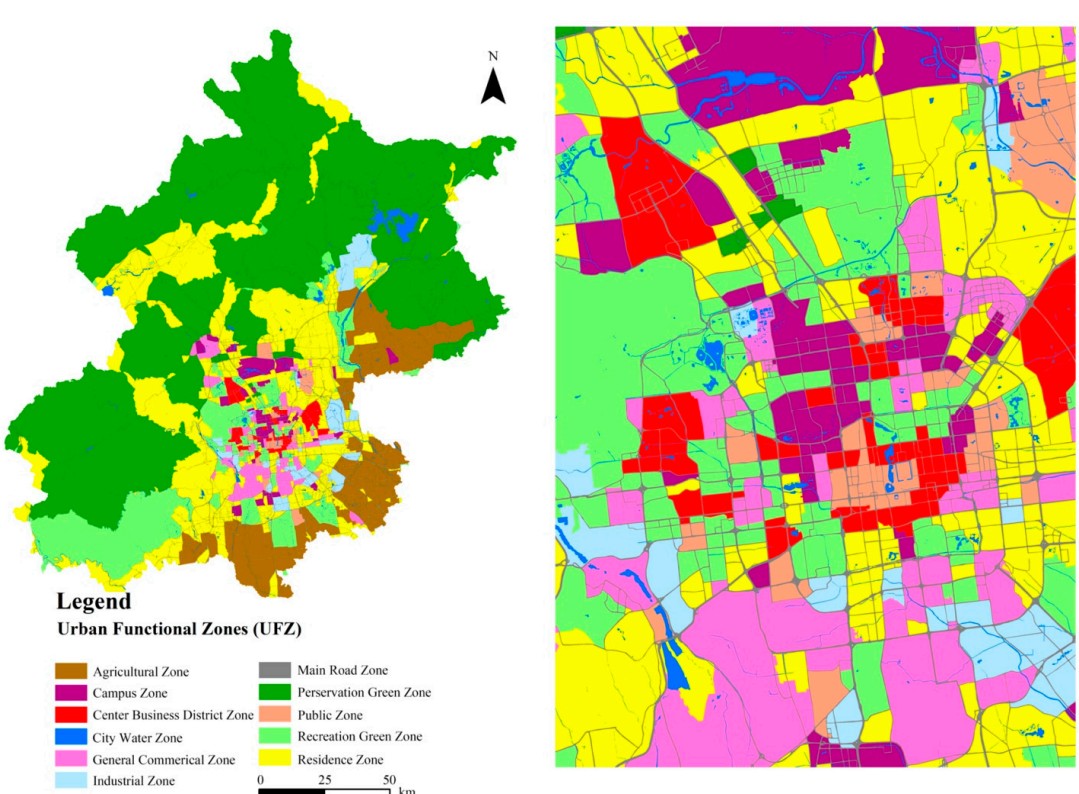

**Figure 5.** Urban functional zone (**left**) and an enlarged view inside the 5th ring-road of Beijing (**right**).

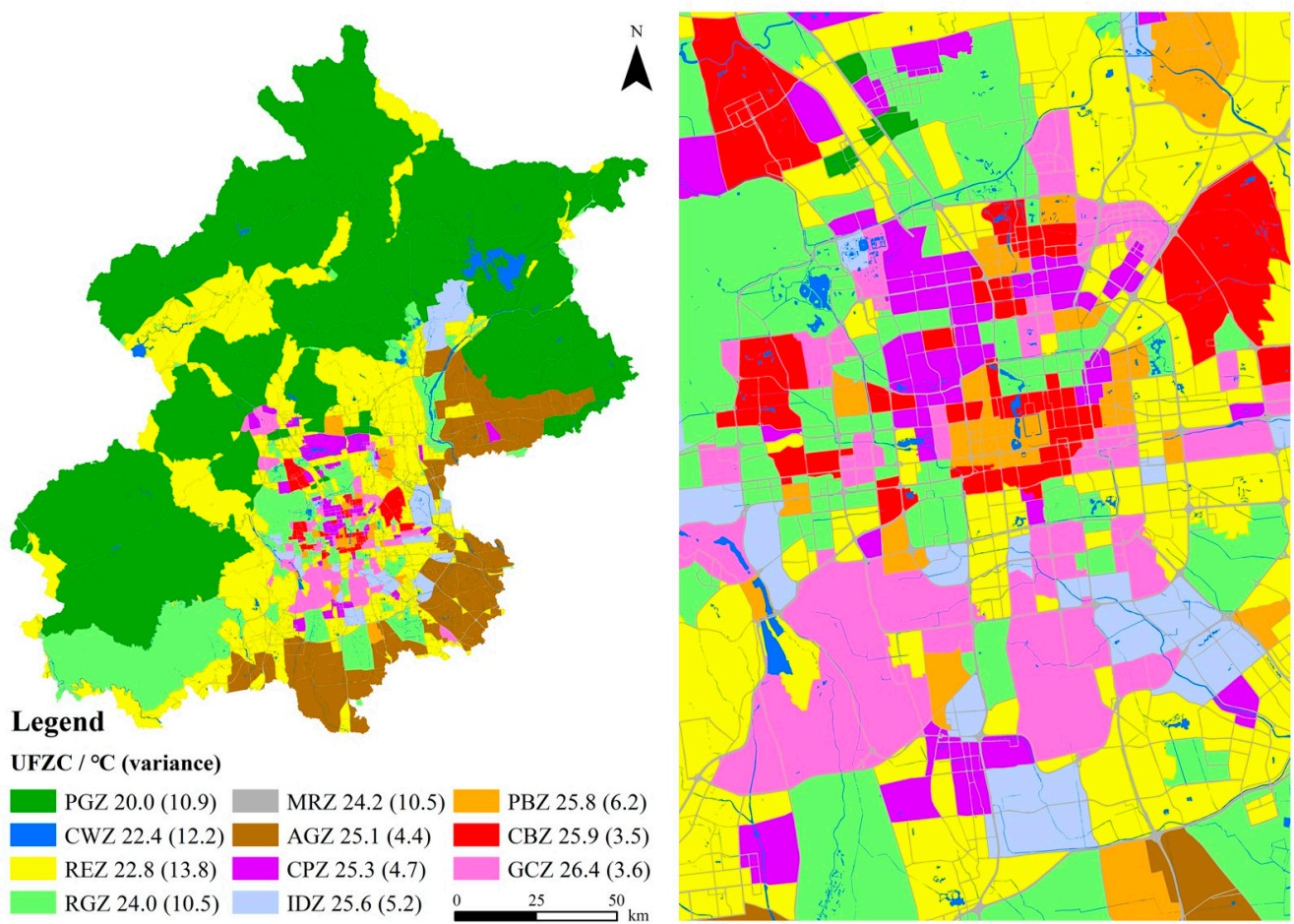

**Figure 6.** The UFZC result of Beijing (**left**) and an enlarged view inside the 5th ring-road of Beijing (**right**).

### 3.3. Results

Results show that preservation green zone (PGZ)-UFZC occupies the largest area of Beijing and is mainly located in the north and west of the Beijing metropolitan area (Figure 6). Besides, the mean LST-based temperature (20.0 °C) of PGZ-UFZC is also the lowest compared to other UFZCs. While the variance of PGZ-UFZC is 10.9 °C, which means that the temperature difference between PGZs is relatively large; and it can be explained in different terrains. The second largest one is REZ-UFZC and is mainly located in the center of Beijing with the mean LST of 22.8 °C and the highest variance (13.8 °C). The agricultural zone (AGZ)-UFZC and industrial zone (IDZ)-UFZC are mainly located in the east and southeast part of Beijing and with mean LST of 25.1 °C (the variance is 4.4 °C) and 25.6 °C (the variance is 5.2 °C) respectively, which means that AGZ-UFZC and IDZ-UFZC are stable heat sources. More interestingly, Figure 6 shows that the commercial activities induced by the center business district zone (CBZ)-UFZC (the variance is 3.5 °C) and general commercial zone (GCZ)-UFZC (the variance is 3.6 °C) have the highest mean LST and lowest variance, which indicates that the commercial activities of Beijing contribute the most and stable heat source. Furthermore, it also implies that reducing the heat generated by the function of commercial activities (and public zone (PBZ), IDZ, campus zone (CPZ)) is an effective measure to reduce the UHI effect.

## 4. Discussion

### *4.1. Advantages of UFZC Classification*

#### 4.1.1. Theoretical Comparison

It is well known that the heterogeneity of the urban internal thermal environment (i.e., canopy layer air temperature) is much greater than that of the urban–rural gradient [1,5]. This heterogeneity is driven not only by land-cover types but also by land-use patterns and human activities [2,30,53,58]. More advanced than UR classification, the LCZ system was developed to some extent to meet this need. The anthropogenic heat caused by land-use patterns and human activities has gradually been recognized as one of the important contributors to urban temperature and the UHI effect, especially in the case of a dense urban agglomeration and high-density neighborhoods [30,31]. The LCZ system, from this point of view, cannot adequately reflect this situation due to the lack of detailed internal (heat) information on the cities.

In general, the RU classification is a large-scale view in an urban–rural gradient region [59], while the LCZ system is a medium-scale in an urban region [17], and the UFZC system usually indicates a small-scale within a city (Figure 7). These can be briefly considered as the different stages of understanding of the urban climate (UHI) classification. The basic principle of the LCZ and UR system emerges from the logical division of the landscape universe, which means that the LCZ and UR system mainly focuses on the physical features of the urban landscape. The logic of the LCZ and UR system is still a physical-based process, and hence, these can be extracted by remote-sensing images to compare and communicate on a large scale [16,60,61]. Cities are complex, diverse, and contain many human activities. Following the logic and methods of study of the natural sciences, it is difficult to obtain comprehensive and effective information in urban-related studies. In urban geography, for instance, the traditional (physical-based) methodology cannot fully understand human behavior characteristics (i.e., social ties, human activities and movements, perception, and cognition). Therefore, as we mentioned above, the new concept of social sensing (SS) was proposed to map the spatiotemporal patterns of human behaviors, and consequently, to reveal socio-economic geographical features [35]. Hence, urban temperature studies also need to transcend the traditional physical-based research process in natural science. The logic and method should be a social-sensing-based process combined with big data, cloud computing, remote sensing, temperature observation, and numerical simulation; which is also the main aim of the UFZC system. Generally, the logic and methods of the UFZC system transcend the traditional physical-based research process in urban temperature study. It combines the physical and non-physical properties of the urban components, as well as integrates the social-sensing and remote-sensing methodologies (Figures 1 and 7).

Therefore, theoretically, the basic logic and hypothesis of the UR and LCZ classification that considers cities as physics-based subjects face challenges in urban temperature research; while the UFZC system (social-based) transcends traditional views and shifts from appearance to the interior, which may be a new approach to urban temperature research. It needs to be emphasis is that although UFZC is a social-based framework, it does not mean the system did not consider the physical property; it combined both two of these, the difference is the UFZC system was initially from the social property (or social-based).

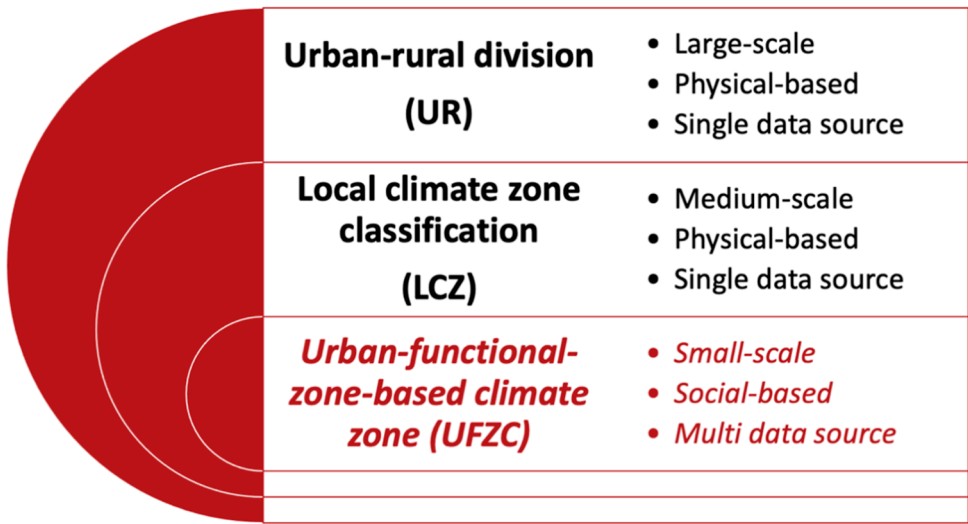

**Figure 7.** A brief overview of the understanding of the urban temperature (and urban heat island (UHI)) classification and comparison of urban–rural (UR), local climate zone (LCZ), and UFZC system.

### 4.1.2. Technical Comparison

The LCZ and UR system comes from traditional urban climate studies, which include observing and documenting temperature datasets, as well as numerical simulation and recent developed remote sensing retrieved data [15,16,19,62]. As mentioned above, UFZ is characterized by similar spectral (physical) features, socio-economic function, and structured by a specific function [33]. It, therefore, has a similar energy consumption and the outdoor (thermal) environment [32]. Hence, within a city, the UFZ-based climate zone (UFZC), combined with physical (spectral) properties and non-physical (social and economic) properties, can provide more accurate information than the LCZ system. For instance, from the view of data sources, the LCZ and UR system is a single data source (e.g., weather station data, mobile car data, and RS data), which means that one data source is enough to obtain LCZ and RU system. However, in a UFZC classification, multiple (at least two—socioeconomic based UFZ data and corresponding temperature data) data sources need to be included (Figure 7).

The technical limitation of the LCZ and UR system is overcome by the UFZC system. The latter integrates Remote Sensing and Social Sensing with a combination of big data and cloud computing (Figure 1), which greatly offers the possibilities of integrating socioeconomic data in urban temperature studies. Therefore, the UFZC system can significantly recognize the effects of anthropogenic heat emissions in different UFZ on the urban temperature compared to the LCZ and UR system.

### 4.1.3. Application Comparison

It is clear that compared to the RU division, in addition to climate modeling and temperature analysis, the LCZ system is more conducive to the analysis and less prone to confusion regarding the UHI magnitude [14]. Nonetheless, the integration of urban climate knowledge with city management has not been especially useful due to the slow advances around the issues of scale and communication [2,14]. The limitations of the LCZ (and UR) system are seen when it is applied to two of the most important aspects of city management—urban planning and energy saving.

Regarding urban planning (or energy saving), the basic planning unit is the urban functional zone (UFZ), i.e., industrial zone, residential zone, agricultural zone, commercial zone, and road [30,33]. In particular, previous studies have demonstrated that temporal-spatial patterns of urban temperature are correlated with the anthropogenic heat (social-economic activities) within the city; and human activities are highly correlated with UFZ [11,32]. The LCZ (and UR) system, on the one hand, is based on the physical properties of the urban

landscape and, therefore, cannot be in line with the basic planning unit—UFZ. On the other hand, the use of UFZ can provide more accurate information than the use of land use and cover [30]. Therefore, the results of LCZ cannot offer a consistent mapping to urban planning and energy saving within a city, and this will make it difficult to apply the LCZ results to city management practices. This is one of the most obvious shortcomings of the LCZ system, although the LCZ system can also guide urban planning to some extent.

Moreover, from the view of city management practice, the order is from UFZ needs to land-use and -cover pattern control [34]; yet the logic of the LCZ system is the opposite. This is another difference between the LCZ and UFZC systems, and it also shows the UFZC system is a good means of integrating urban management.

### 4.2. Limitations of UFZC System and Further Studies

Firstly, the UFZC system integrates big data, cloud computing, and remote-sensing technology, especially corresponding computer programs; hence the use of UFZC systems in other cities requires a certain learning cost. Nevertheless, we have already published the relevant program code (Supplemental Materials A and B), so if other researchers want to carry out the related research, they can directly use the code provided by us, which will avoid certain learning costs. Secondly, the UFZC classification itself has some challenges. For instance, the mean LST of residence zones (REZ)-UFZC in Beijing is 22.8 °C, and the variance is 13.8 °C; this is also the UFZ with the largest temperature change among the 11 UFZ types. When we delve into the reasons for this result, we can find that the types of residential area are diverse in a city, such as high-density residential areas in urban centers, relatively low-density residential areas in suburbs, and low-density residential areas in the outer suburbs. This pattern is likely to be an important cause of large temperature differences in REZ-UFZC type; this also reminds us that the next study can further extract more detailed data and form sub-types. Thirdly, the case study used LST data to represent the corresponding UFZ temperature due to the limitations of air temperature data acquisition in different UFZs, which is another challenge in this case study. For example, we speculate that if the air temperature (rather than LST) is used to represent urban temperature, then the temperature of UFZ dominated by human activities should be higher. In other words, the temperature of the Beijing cases such as main road zone (MRZ)-UFZC, GCZ-UFZC, and CBZ-UFZC are underestimated. In the next study, we can use more high-resolution LST data or obtain more accurate air temperature data to make the UFZC results more accurate, thus providing a more direct basis for urban climate alleviation.

Furthermore, with the development of artificial intelligence (AI), it has been reported that machine-learning methods have the potential to improve the accuracy of training samples and classification [33]. Hence, in the next research, we hold the opinion that machine-learning methods can be applied to train the weights of POIs and functional segmentation. In addition to that, we also suggested that the definition and explanation of UHI should be based on UFZCs rather than UR differences or LCZ differences.

### 5. Conclusions

Accurately defining and identifying UHI is an important step in mitigating the UHI effect. In this study, the integration of social sensing and remote sensing, we developed a new urban functional zone-based (UFZC) urban temperature zoning system. Through a comparison (theory, technology, and application) of the previous definition (UR and LCZ) of UHI, we suggested that the new concept of UFZC can be a better classification system for urban temperature study due to the high probability of obtaining detailed physical and non-physical (human activities) information. We think the UFZC system is generally a social-based, planning-oriented, and data-driven classification system associated with the urban function and temperature. Moreover, to test the effectiveness of this classification, we chose Beijing as a case for analysis and we revealed patterns and causes of 11 UFZCs in the Beijing metropolitan. Specifically, results show that the PGZ-UFZC has the lowest LST, while the

CBZ-UFZC and GCZ-UFZC contribute the most and stable heat sources, which implies that reducing the heat generated by the function of commercial (and industrial) activities is an effective measure to reduce the UHI effect. In addition to the value of the study case area, we believe that the more important value of this study is that we can apply this method and UFZC classification system to other cities to accurately locate the UFZC-based UHI.

**Supplementary Materials:** The following are available online at https://www.mdpi.com/2072-429 2/13/2/251/s1, Supplemental Material A: Python Programming Code. Supplemental Material B: Online map data crawl system for urban land use data with functional properties.

**Author Contributions:** Conceptualization, Z.Y., G.Y., R.S. and methodology, Z.Y. and Y.J.; processing and analyses of data: Z.Y., Y.J., visualization, Z.Y., Y.J.; writing—original draft preparation, Z.Y., G.Y. and Y.J.; writing—review and editing, Z.Y., G.Y. and R.S.; supervision, Z.Y. and R.S. All authors have read and agreed to the published version of the manuscript.

**Funding:** This research was funded by the National Natural Science Foundation of China (grant no. 41922007), the Open Foundation of the State Key Laboratory of Urban and Regional Ecology of China (grant no. SKLURE2019-2-6).

**Institutional Review Board Statement:** Not applicable.

**Informed Consent Statement:** Not applicable.

**Data Availability Statement:** The data presented in this study are available on request from the corresponding author.

**Acknowledgments:** We also thanks anonymous reviewers and academic editor for their constructive comments and suggestions.

**Conflicts of Interest:** The authors declare no conflict of interest.

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
