# Peer review of "A New Urban Functional Zone-Based Climate Zoning System for Urban Temperature Study"

_remotesensing, doi:10.3390/rs13020251_

Round 1

Reviewer 1 Report

The abstract section must define in a clear way the main gap of knowledge, how is addressed by the paper and what is the main conclusion.

Introduction section:

Please revise the average height of the canopy layer. It is not 1-2 m above ground.

Anthropogenic heat is mentioned as the one reason for urban climate, but this is not the only reason. There are many modifications of the urban sit in comparison with the rural environment to justify the differences causing the UHI. Not just a question of human activities.

Introduction section should end with a clear message of scientific novelty and not only of definition of new terms.

Methodology section:

References are needed for most of the software tools that are mentioned.

The same applies to different methodologies or concepts. For instance: Entropy Method.

It is common to find in the scientific literature the use of the base-2 logarithm to express the probability of an event. Please, justify the equations and make a reference.

At the end of section 2:

How the land surface emissivity is calculated as a function of the water, artificial and nature emissivities?.

An explanation on the need for these equations is missing. It is not clear the purpose and how is used for the UFZC classification.

Case study section:

In my opinion you also need a map with building heights and building density, traffic information, soil type, colours, sky view factor, etc… since this information is crucial for the understanding of UHI. Please, justify why it is not considered.

Please explain how to understand the UHI causes without a physical based model, and how to propose measures to improve it.

Conclusion section:

The conclusion section need to clearly express what is new and the interest of this achievement in comparison with the state of art.

Author Response

Here is not very clear, please read our attached file.

Reviewer 1

The abstract section must define in a clear way the main gap of knowledge, how is addressed by the paper and what is the main conclusion.

 Answer: thank you for your comments.

We have made some changes to the manuscript. Here we explain to you the logic of the Abstract.

First, we stated the general information why we did this study, that is: “Urban heat island (UHI) effect has been recognized as one of the most significant terrestrial surfaces climate-related consequences of urbanization.”

Second, we described the gaps of the UHI definition, that is: “However, the traditional definition of urban-rural (UR) division and the newly established local climate zone (LCZ) classification for UHI and urban climate studies do not adequately express the pattern and intensity of UHI. Moreover, it is difficult to quantify the underlying drivers of UHI caused by human activities that are highly correlated with urban functional zones (UFZ).”

Third, regarding these gaps, we proposed our new system and explained why this definition is meaningful, that is: “Therefore, in this paper, with the comparison (theory, technology, and application) of the previous definition (UR and LCZ) of UHI and integration of computer programming technology, social sensing, and remote sensing, we developed a new urban-functional-zone-based urban temperature zoning system (UFZC). The UFZC system is generally a social-based, planning-oriented, and a data-driven classification system associated with the urban function and temperature; it also can be effectively used in city management (e.g., urban planning and energy saving).

Fourth, we used Beijing case a case to test the system, and we found results and stated as follows: “In addition, in Beijing case, we tested the UFZC system and preliminary analyzed the land surface temperature (LST) difference patterns and causes of the 11 UFZC types. We found that compared to other UFZCs, PGZ (Perseveration Green Zone) -UFZC has the lowest LST, while the CBZ (Center Business District Zone)-UFZC and GCZ (General Commercial Zone)-UFZC contribute the most and stable heat source. Which implies that reducing the heat generated by the function of commercial (and industrial) activities is an effective measure to reduce the UHI effect. We also proposed that multi-source temperature datasets with the high spatiotemporal resolution are needed to get more accurate results; thus providing more accurate recommendations for mitigating UHI effects.”

Finally, we conclude that: “In short, as a new and finer urban temperature zoning system, although UFZC is not intended to supplant the UR and LCZ classifications, it can facilitate more detailed and coupled urban climate studies.”

We think the write logic of the manuscript is clear now.

If you have any comments, please let us know.

Thank you again.

Introduction section:

Please revise the average height of the canopy layer. It is not 1-2 m above ground.

Answer: thank you for your comments.

We have checked the Book Urban Climate by Oke (2017). We have changed the sentence as follows:

In general, the quantification of UHI is related to the urban canopy layer, which is observed using thermometers to measure air temperature (Ta) near the ground (Ta can be measured in a weather screen or ventilated radiation shield, at one or more sites considered to be representative of urban and rural); air in this layer is typically warmer than that at screen height in the countryside [1].”

Anthropogenic heat is mentioned as the one reason for urban climate, but this is not the only reason. There are many modifications of the urban sit in comparison with the rural environment to justify the differences causing the UHI. Not just a question of human activities.

Answer: thank you for your comments.

That is right!!! we agree with your comments that Anthropogenic heat is not the only reason for UHI. This is one of the contributions of the study that we highlight the effect of anthropogenic heat on the UHI effect, while previous definition RU and LCZ did not consider it. 

In our system, we not only consider natural factors (land cover and use) but also further consider anthropogenic heat emission. We integrated social and physical factors.

We think the figure will make you clearly understand the advantages of our system.

 We hope we have answered your concerns.

Please let us know if you have any questions.

Introduction section should end with a clear message of scientific novelty and not only of definition of new terms.

Answer: thank you for your comments.

We agree with your comments and we have rewritten the sentence to make it clear.

Therefore, to address the abovementioned insufficiencies and provide a general microscale UHI definition system, the study aims to: (1) propose a new method to mapping UFZ with the employment of RS and SS technology; and (2) further put forward a new urban-functional-zone-based urban temperature (UFZC) zoning system for urban temperature studies; as well as the advantages (and limitations) of UFZC classification with previous classification schemas (UR and LCZ schema). The proposed UFZ mapping method and particularly urban temperature zoning system – UFZC would deepen our understanding of the urban thermal environment at a finer scale.

Thank you again for your constructive comments.

Methodology section:

References are needed for most of the software tools that are mentioned.

The same applies to different methodologies or concepts. For instance: Entropy Method.

 Answer: thank you for your comments.

We agree with your comments that software tools should be describe in detailed and we have updated the references of the manuscript.

  1. Delgado, A.; Romero, I. Environmental conflict analysis using an integrated grey clustering and entropy-weight method: A case study of a mining project in Peru. Environmental Modelling & Software 2016, 77, 108-121, doi:https://doi.org/10.1016/j.envsoft.2015.12.011.
  2. Zou, Z.-h.; Yun, Y.; Sun, J.-n. Entropy method for determination of the weight of evaluating indicators in fuzzy synthetic evaluation for water quality assessment. Journal of Environmental Sciences 2006, 18, 1020-1023, doi:10.1016/s1001-0742(06)60032-6.

ArcGIS 10.2, Python, and PyCharm are the software or tools we used in this study. There are some of the core packages: NumPy, SciPy, pandas, matplotlib which had been widely used. And the Entropy Method was also done based on the Entropy Method equations in Python. However, the software or tools not the critical part of this paper, other tools, like R or MATLAB, can implement the same results, so we didn’t detail in the manuscript. Of course, we will publish codes and detail the python packages if the paper is accepted by the journal.

Thank you again for your constructive comments.

Please let us know if you have other questions.

It is common to find in the scientific literature the use of the base-2 logarithm to express the probability of an event. Please, justify the equations and make a reference.

 Answer: thank you for your comments.

We agree with your comments that the base-2 logarithm is largely used in the scientific literature. But base-e logarithm is used in the Entropy Method not base-2 logarithm. The Cross-entropy method was developed by Reuven Rubinstein (1999), and is widely used in many scientific studies. We have made some changes to the manuscript and updated the references as follows:

  1. Rubinstein, R. The Cross-Entropy Method for Combinatorial and Continuous Optimization. Methodology And Computing In Applied Probability 1999, 1, 127-190, doi:10.1023/a:1010091220143.

At the end of section 2:

How the land surface emissivity is calculated as a function of the water, artificial and nature emissivities?.

 Answer: thank you for your comments.

We agree with your comments and we have rewritten the sentence to make it clear. The land surface emissivity was calculated based on the single-channel method which needs the indicator FVC (fractional vegetation cover), and the equation is an empirical formula wildly used. We detailed the equation of FVC and update the references of the dimidiate pixel model to calculate FVC. The dimidiate pixel model has a fairly sound theoretical basis and is widely applicable regardless of the geographical constraints.

The updated references as follows

  1. Carlson, T.N.; Ripley, D.A. On the relation between NDVI, fractional vegetation cover, and leaf area index. Remote Sensing of Environment 1997, 62, 241-252, doi:10.1016/s0034-4257(97)00104-1.
  2. Vanhellemont, Q. Combined land surface emissivity and temperature estimation from Landsat 8 OLI and TIRS. ISPRS Journal of Photogrammetry and Remote Sensing 2020, 166, 390-402, doi:10.1016/j.isprsjprs.2020.06.007.

  1. Carlson, T.N.; Ripley, D.A. On the relation between NDVI, fractional vegetation cover, and leaf area index. Remote Sensing of Environment 1997, 62, 241-252, doi:10.1016/s0034-4257(97)00104-1.
  2. Vanhellemont, Q. Combined land surface emissivity and temperature estimation from Landsat 8 OLI and TIRS. ISPRS Journal of Photogrammetry and Remote Sensing 2020, 166, 390-402, doi:10.1016/j.isprsjprs.2020.06.007.

An explanation on the need for these equations is missing. It is not clear the purpose and how is used for the UFZC classification.

Answer: thank you for your comments.

As shown in Figure 1the technical framework of UFZC system identification, the aim of section 2.2 is to get UFZ of the study area (in the green box), the aim of section 2.3 is to get LST of the study area (in the red box), “Finally, after the above process, we can perform UFZC classification in any target area (Beijing).” We hope this figure could make it clear.

Case study section:

In my opinion you also need a map with building heights and building density, traffic information, soil type, colours, sky view factor, etc.… since this information is crucial for the understanding of UHI. Please, justify why it is not considered.

Answer: thank you for your comments.

We agree with your comment that if you want to accurately map the UHI intensity, the factors such as building heights and building density, traffic information, soil type, colors, and sky view factor need to be considered.

However, this is not the main aim of this paper. As you can see, the study aims to propose an urban-functional-zone-based urban temperature zoning system.

Regarding the Beijing case, the primary results have limitations we have stated: “Besides, it needs to be mentioned that the result of UFZC classification in Beijing is only using one-snap Landsat image (the image was captured on September 28, 2018), hence the result of this test may not so accurate, although we can still see the LST difference in different UFZCs generally (the results are similar to previous studies by Sun et al. [29]. More importantly, in the next study, multi-source temperature data sets with the high temporal and spatial resolution are needed to get more accurate results; thus, providing more accurate recommendations for mitigating UHI effects.

In section 4.2 Limitations of UFZC, we also stated the limitations of the study.

“Thirdly, the case study used LST data to represent the corresponding UFZ temperature due to the limitations of air temperature data acquisition in different UFZs, which is another challenge in this case study. For example, we speculate that if the air temperature (rather than LST) is used to represent urban temperature, then the temperature of UFZ dominated by human activities should be higher. In other words, the temperature of the Beijing case such as MRZ-UFZC, GCZ-UFZC, and CBZ-UFZC is underestimated. In the next study, we can use more high-resolution LST data or obtain more accurate air temperature data to make the UFZC results more accurate, thus providing a more direct basis for urban climate alleviation.”

We think we have described the limitations of the study.

We hope the answer can satisfy you and please let us know if you have more questions.

Please explain how to understand the UHI causes without a physical based model, and how to propose measures to improve it.

Answer: thank you for your comments.

I’m not sure if my guess is accurate, but we think you might have misunderstood our meaning of the UFZC. In the manuscript, we have described that the UFZC is a social-based framework, but it doesn’t mean that we did not consider the physical property. We combined both two of these, the difference is we initial from the social property (or social-based). As we described:

“The logic and method should be a social-sensing-based process combined with big data, cloud computing, remote sensing, temperature observation, and numerical simulation; which is also the main aims of the UFZC system. Generally speaking, the logic and methods of the UFZC system transcend the traditional physical-based research process in urban temperature study. It combines the physical and non-physical properties of the urban components, as well as integrates the social sensing and remote sensing methodologies (Figure 1).”

So we do not think it is a problem.

Therefore, as we found in the Beijing case, business and industrial activities are two main factors that influence the surface UHI, while green space is the heat sink, so measures need to implement in these two parts (but we will not discuss this issue, because it is not the main aims of the study).  

Conclusion section:

The conclusion section need to clearly express what is new and the interest of this achievement in comparison with the state of art.

Answer: thank you for your comments.

We agree with your comments, and we have rewritten the conclusion part.

Please see the follows:

Accurately defining and identifying UHI is an important step in mitigating the UHI effect. In this study, the integration of social sensing and remote sensing, we developed a new urban-functional-zone-based (UFZC) urban temperature zoning system. With the comparison (theory, technology, and application) of the previous definition (UR and LCZ) of UHI, we suggested that the new concept of UFZC can be a better classification system for urban temperature study due to the high probability of obtaining detailed physical and non-physical (human activities) information. Hence we think the UFZC system is generally a social-based, planning-oriented, and data-driven classification system associated with the urban function and temperature. Besides, to test the effectiveness of this classification, we chose Beijing as a case for analysis and we revealed patterns and causes of 11 UFZCs in the Beijing metropolitan. Specifically, results show that the PGZ-UFZC has the lowest LST, while the CBZ-UFZC and GCZ-UFZC contribute the most and stable heat source, which implies that reducing the heat generated by the function of commercial (and industrial) activities is an effective measure to reduce the UHI effect. In addition to the value of the study case area, we believe that the more important value of this study is that we can apply this method and UFZC classification system to other cities to accurately locate the UFZC-based UHI.

Thank you again for your constructive comments.

Reviewer 2 Report

The proposed study is interesting and original since it deals with a key-question in addressing the issue of the definition of urban zoning taxonomies related to urban heat islands (UHIs). The manuscript defines a methodology to identify a detailed classification of the urban territory named “urban-functional-zone-based urban temperature zoning system” (UFZC) whose implementation should result in a new system of urban climate zones (UCZs) more effective in representing the UHI phenomenon in urban and metropolitan contexts than the present taxonomies concerning UCZs. The methodology is implemented with reference to the Beijing metropolitan context. The new UFZC is based on the integration of remote sensing, related to the estimates of the spatial distribution of urban temperature, and social sensing technologies, which help to identify the impacts of anthropogenic factors on heat emissions.
In my opinion, the study should not be published in its present form, since a number of caveats need to be addressed in a proper way.
In a revised version of the study, the authors should carefully address the following points.
i. Section 1 “Introduction.” Although it is conceivable that the choice of the Beijing metropolitan area as the spatial context to implement the proposed methodology was based on the familiarity of the authors with this spatial context and on data availability, I would recommend the authors discuss a comparison between Beijing and other national and international urban and metropolitan areas in order to make clear the interest of the submitted manuscript for the vast scientific and technical public of the readers of Remote Sensing.
ii. Section 1 “Introduction” and Section 2 “Methodology.” The research questions are given at the end of Section 1, as follows: “1) propose a new urban-functional-zonebased urban temperature (UFZC) zoning system for urban climate studies; and 2) examine the advantages (and limitations) of UFZC classification with previous classification schemas (UR and LCZ schema).” I would recommend the authors, perhaps at the beginning of Section 2, describe the context of the current literature concerning these research questions. I would recommend the authors discuss in depth the research questions and the ways they adopt to address them with reference to available studies related to the definition of urban taxonomies concerning UCZs, explain why these ways were adopted instead of alternative approaches, and, most important, describe which is the value added of the submitted manuscript with reference to the current studies. In particular, as regards the integration of social sensing and remote sensing technologies, some references drawn from the current literature are quoted in Section 1, but their relations to the manuscript ways of addressing the research questions are not clear at all.
iii. Section 4 “Discussion.” As regards subsections 4.1.1. “Theoretical comparison”, 4.1.2. “Technical comparison” and 4.1.3 “Application Comparison,” I would recommend the authors add a detailed comparative analysis of the results, perhaps by adding a fourth subsection “4.1.4.”, with reference to the studies concerning UCZs taxonomies available in the international literature, the research gaps they would like to address through the submitted manuscript and how these gaps are filled through the outcomes obtained through their study. In my opinion, the present version of the discussion is focused on the Beijing case study whereas generalization is missing. Moreover, I would recommend the authors analytically discuss the advancements implied by their manuscript as compared to the current literature, in order to make the reader aware of the value added of the submitted article.
iv. Section 5 “Conclusions.” I would recommend the authors be specific in discussing the implications of the outcomes of the study as regards the exportability of the implemented assessment to international contexts different from the Beijing metropolitan context. In other words, I would recommend the authors make the reader aware of the reasons the submitted manuscript is likely to be helpful in defining new and more effective UFZCs in other countries’ urban and metropolitan contexts.

Author Response

Here is not very clear, please read our attached file.

Reviewer 2

The proposed study is interesting and original since it deals with a key-question in addressing the issue of the definition of urban zoning taxonomies related to urban heat islands (UHIs). The manuscript defines a methodology to identify a detailed classification of the urban territory named “urban-functional-zone-based urban temperature zoning system” (UFZC) whose implementation should result in a new system of urban climate zones (UCZs) more effective in representing the UHI phenomenon in urban and metropolitan contexts than the present taxonomies concerning UCZs. The methodology is implemented with reference to the Beijing metropolitan context. The new UFZC is based on the integration of remote sensing, related to the estimates of the spatial distribution of urban temperature, and social sensing technologies, which help to identify the impacts of anthropogenic factors on heat emissions.

In my opinion, the study should not be published in its present form, since a number of caveats need to be addressed in a proper way.

In a revised version of the study, the authors should carefully address the following points.

  1. Section 1 “Introduction.” Although it is conceivable that the choice of the Beijing metropolitan area as the spatial context to implement the proposed methodology was based on the familiarity of the authors with this spatial context and on data availability, I would recommend the authors discuss a comparison between Beijing and other national and international urban and metropolitan areas in order to make clear the interest of the submitted manuscript for the vast scientific and technical public of the readers of Remote Sensing.

Answer: thank you for your comments.

Firstly, what we want to emphasize is that the main aims of the study are to propose a method to map UFZ and especially propose a new urban-functional-zone-based Urban Temperature Zoning System for Urban Temperature Study. Therefore, we think the comparison study is not so important. It is also not the main aim of the study.

Secondly, we think the case study (Beijing) already can demonstrate the value of our framework.

Thirdly, Different cities have their unique characteristics, resulting in different final results, which are not comparable.

Fourthly, we will publish all codes and methods if the paper is accepted by the journal. Other researchers can use our methods to analyze other cities.

Finally, we think your comment is very similar to our next study. Actually, in the future study, what we want to do is analyze the typical and similar cities UHI patter based on UFZC.

We hope we have answered your comments.

Please let us know if you have other questions.

  1. Section 1 “Introduction” and Section 2 “Methodology.” The research questions are given at the end of Section 1, as follows: “1) propose a new urban-functional-zonebased urban temperature (UFZC) zoning system for urban climate studies; and 2) examine the advantages (and limitations) of UFZC classification with previous classification schemas (UR and LCZ schema).” I would recommend the authors, perhaps at the beginning of Section 2, describe the context of the current literature concerning these research questions. I would recommend the authors discuss in depth the research questions and the ways they adopt to address them with reference to available studies related to the definition of urban taxonomies concerning UCZs, explain why these ways were adopted instead of alternative approaches, and, most important, describe which is the value added of the submitted manuscript with reference to the current studies. In particular, as regards the integration of social sensing and remote sensing technologies, some references drawn from the current literature are quoted in Section 1, but their relations to the manuscript ways of addressing the research questions are not clear at all.

Answer: thank you for your comments.

We agree with your comments and we need to more clearly describe the gaps and research questions in the introduction section.

We have made a significant improvement in the introduction part.

Here is the writing logic in the introduction part.

In the first paragraph, we briefly describe the urbanization and UHI effect.

“The urban climate is one of the hot topics in urban environment research [1]. According to the World Urbanization Prospects 2018, 55% of the world’s population reside as urban dwellers in 2018, and by 2050, that number is expected to reach 68%. Urbanization has transformed the natural surfaces into a coupled human and natural system, which alter the materials, energy flows, radiation, and composition of the atmospheric structure in the near-surface layer [2-5]. During urbanization, the modification of terrestrial surfaces, changes in surface material and the albedo of buildings (pavement), the geometry of the surface structure (e.g. spacing and height of buildings), and anthropogenic heat emissions cause one of the most significant impacts on the Earth’s surface climate – urban heat island (UHI) [6-10].”

Secondly, we detailed described the UHI definition evolution and related classifications to describe the urban climate (temperature) in cities.

In general, the quantification of UHI is related to the urban canopy layer, which is observed using thermometers to measure the air temperature (Ta) near the ground (Ta can be measured in a weather screen or ventilated radiation shield, at one or more sites considered to be representative of urban and rural); air in this layer is typically warmer than that at screen height in the countryside [1]. Hence, the classical definition of urban-rural classification (UR) of UHI is that the atmospheric temperature in the urban area is warmer than its surrounding rural areas [6,11-13]. The UR classification of UHI has given researchers and decision-makers a simple and intuitive framework to separate the urban and rural effects on local climate [14]. However, the classical UR classification has recently suffered critical challenges [15-18]. For instance, Stewart and Oke [14]) suggested that while the definitions of urban and rural may be evocative of the landscape, they are vague as an object of scientific analysis. It also claimed that more than three-quarters of the observational UHI literature fails to give the local or micro-scale character of those sites and rarely reports on the site metadata necessary to quantify or otherwise elucidate the terms ‘urban’ and ‘rural’ [19]. Particularly, the classical UR classification is becoming outmoded today, especially in the densely populated (i.e., Asia) developing urban agglomeration region [12]. It also mentioned that the relationship between urban and rural is a dynamic and continuous process rather than a dichotomy [20], while the spatial demarcation between the urban and rural is normally artificial, the term urban has no single, objective meaning and thus has no climatological relevance as Stewart and Oke [14] stated. Therefore, they proposed a new classification schema (local climate zone, LCZ) to facilitate consistent and climatologically relevant studies and classifications. The schema includes 17 standard classes at the local scale (102 to 104 m), and each class is unique in its combination of surface structure, fabric, and metabolism (to some extent). Moreover, the previous research [12] also indicated that classical UR classification is not suitable to examine the heat effect in the context of urban agglomerations, and the concept of UHI should be replaced by region heat island (RHI). In addition, there also have some other classifications to describe urban climate (temperature) in cities like urban terrain zones (UTZs) [21], urban climate zones (UCZs) [2], urban zones for characterizing energy partitioning (UZEs) [22], and Climatopes [23].”

Thirdly, we further described that cities are a highly-populated area with various socioeconomic activities. And emphasis the human activities induced anthropogenic heat to contribute a direct external source to the urban thermal environment. Subsequently, we introduced the UFZ and suggested that the UFZ is normally characterized by similar spectral features, socio-economic function, and structured by a specific function and therefore has a similar energy consumption and the outdoor thermal environment.

Fourthly, although we proposed that the UFZ is meaningful to urban temperature studies, the accuracy of UFZ mapping is limited. Therefore, we stated that the studies combining multi-source data, such as RS and SS (social sensing) data to mapping UFZ is necessary.

Finally, we proposed that based on the above insufficiencies and provide a general microscale UHI definition system, the study aims to:

“(1) propose a new method to mapping UFZ with the employment of RS and SS technology; and (2) further put forward a new urban-functional-zone-based urban temperature (UFZC) zoning system for urban temperature studies; as well as the advantages (and limitations) of UFZC classification with previous classification schemas (UR and LCZ schema).”

We think the writing logic is clear and readable.

We hope we have answered your questions.

Please let us know if you have any questions.

iii. Section 4 “Discussion.” As regards subsections 4.1.1. “Theoretical comparison”, 4.1.2. “Technical comparison” and 4.1.3 “Application Comparison,” I would recommend the authors add a detailed comparative analysis of the results, perhaps by adding a fourth subsection “4.1.4.”, with reference to the studies concerning UCZs taxonomies available in the international literature, the research gaps they would like to address through the submitted manuscript and how these gaps are filled through the outcomes obtained through their study. In my opinion, the present version of the discussion is focused on the Beijing case study whereas generalization is missing. Moreover, I would recommend the authors analytically discuss the advancements implied by their manuscript as compared to the current literature, in order to make the reader aware of the value added of the submitted article.

Answer: thank you for your comments.

Firstly, As we answered above, we think the most important message we want to deliver in this study is the UFZC classification system. It is not about the results of the case study; the case study just wants to show the value of the UFZC system. We also can see the Beijing case proved our hypothesis, such as we stated “More interestingly, Figure 6 shows that the commercial activities induced CBZ-UFZC (the variance is 3.5℃) and GCZ-UFZC (the variance is 3.6℃) have the highest mean LST and lowest variance, which indicates that the commercial activities of Beijing contribute the most and stable heat source. Further, it also implies that reducing the heat generated by the function of commercial activities (and PBZ, IDZ, CPZ) is an effective measure to reduce the UHI effect.”

Secondly, we think the subsection 4.1.1, 4.1.2, and 4.1.3 have already answered the difference of UR, LCZ, and UFZC system. Particularly, in this discussion, we referenced many international publications, and we think the discussion clearly showed the advantages of the UFZC system.

Thirdly, the logic of the study is as follows: (1) we firstly proposed a UFZC system; (2) we secondly used Beijing as the case to verify the feasibility of the classification system. (3) we further discuss the advantages of the system. From the basic logic, we can that the present version of the discussion is not focused on the Beijing case; what we want to emphasize is the generalization of the classification. If you read section 4.1, we did not mention the information on the Beijing case. We only mentioned the Beijing case in Section 4.2 limitation of UFZC.

Fourthly, we think the value of the study is clear and we have described above. It references two research questions: (1) we proposed a method to map UFZ accuracy. (2) we proposed a New urban-functional-zone-based Urban Temperature Zoning System for Urban Temperature Study. Based on our comprehensive analysis and discussion (Theoretical comparison, Technical comparison, and Application Comparison), we believe the UFZC system is a promising classification to do urban climate study.

Finally, as we emphasized, we believe that the system is not only a conceptual framework but also a practical generalized system.  All codes will be published when the paper was accepted by the journal, which means researchers can download the codes and use the method to analyze other cities.

We hope we have answered your questions.

Please let us know if you have more questions.

  1. Section 5 “Conclusions.” I would recommend the authors be specific in discussing the implications of the outcomes of the study as regards the exportability of the implemented assessment to international contexts different from the Beijing metropolitan context. In other words, I would recommend the authors make the reader aware of the reasons the submitted manuscript is likely to help define new and more effective UFZCs in other countries’ urban and metropolitan contexts.

Answer: thank you for your comments.

As I mentioned above, the methods (codes) and the UFZC system can be used in other cities. We will publish all program codes when the study is accepted by the journal, which means other researchers no need to waste time writing code.

According to your comments, we have rewritten the conclusion part.

Accurately defining and identifying UHI is an important step in mitigating the UHI effect. In this study, the integration of social sensing and remote sensing, we developed a new urban-functional-zone-based (UFZC) urban temperature zoning system. With the comparison (theory, technology, and application) of the previous definition (UR and LCZ) of UHI, we suggested that the new concept of UFZC can be a better classification system for urban temperature study due to the high probability of obtaining detailed physical and non-physical (human activities) information. We think the UFZC system is generally a social-based, planning-oriented, and data-driven classification system associated with the urban function and temperature. Besides, to test the effectiveness of this classification, we chose Beijing as a case for analysis and we revealed patterns and causes of 11 UFZCs in the Beijing metropolitan. Specifically, results show that the PGZ-UFZC has the lowest LST, while the CBZ-UFZC and GCZ-UFZC contribute the most and stable heat source, which implies that reducing the heat generated by the function of commercial (and industrial) activities is an effective measure to reduce the UHI effect. In addition to the value of the study case area, we believe that the more important value of this study is that we can apply this method and UFZC classification system to other cities to accurately locate the UFZC-based UHI.”

Thank you again for your constructive comments.

Round 2

Reviewer 1 Report

1)

In the abstract section, and then in the main text of the paper the author say "Moreover, it is difficult to quantify the underlying drivers of UHI caused by human activities that are highly correlated with urban functional zones (UFZ)"

The reader gets an impression that here we will find how to quantify UHI drivers. But nothing in this sense is explained in the paper. In fact, the paper doesn't present, like in many other papers and books (see Oke Urban Canopy Layer) the main UHI drivers: reduction of the sky view factor, an increase of thermal inertia materials, reduction of green surfaces, etc. It is not only a matter of anthropogenic heat.

In the pdf response from authors, you have included a comparative graph that you probably should include in the text. It is not clear for me if it is only a matter of scale and a difference between physical or social base.

I recommend to include this figure in the introduction section and a detailed explanation comparing the different models. What do you mean with physical vs social base?

2) 

Even when the software tools used to produce the results of the paper are not the critical part of it, in my opinion, authors should make a reference to each of them the first time they are mentioned. (like this: FAQ: What is the correct way to cite an ArcGIS Online basemap? (esri.com)) (see also: Citing ArcGIS Desktop when writing papers? - Geographic Information Systems Stack Exchange)

3)

I still think it is necessary to clearly state which steps of the procedure in Figure 1 are expressed with equations 1-8, and then, what are the previous and following steps.

4)

In my opinion, to conclude that the way to reduce UHI is through the reduction of commercial and industrial activities is not extracted from this study. A more detailed study using for instance CFD simulations, radiation calculations, etc, could highlight other reasons like: reduction of sky view factor, maybe the use of dark colour surfaces, lack of vegetation...etc. In fact, due to the complexity of the system, detailed software tools like ENVIMET can help to that purpose.

In any case, it is not possible to conclude without performing a detailed analysis

Author Response

Detailed response, Please see the attached file. 

Reviewer 2 Report

As mentioned in the author-response cover letter, the authors propose a rebuttal of the first point I raised in the first place. Of course, I do not agree with their position, since I do not think that a manuscript submitted to a high-level international journal can omit discussing the relationship of their study with other relevant international contexts, in order make the reader aware of the general interest of their research. Moreover, in my opinion the following statement is not acceptable as part of the autors' rebuttal: "Finally, we think your comment is very similar to our next study. Actually, in the future study, what we want to do is analyze the typical and similar cities UHI patter based on UFZC", since the authors claim that a future and not-yet-available study will address a caveat of the submitted manuscript.

All the other points I raised in the first place are appropriately addressed in the revised version of the manuscript.

I leave to the Academic Editor of Remote Sensing the task to  evaluate if the article can be accepted for publication in its present version. 

Author Response

Detailed response, please see the attached file.
